# Evaluation of an automated feedback intervention to improve antimicrobial prescribing among primary care physicians (OPEN Stewardship): protocol for an interrupted time-series and usability analysis in Ontario, Canada and Southern Israel

Jean-Paul R Soucy [1] ,[1] Marcelo Low,[2] Kamal Raj Acharya [1] ,[3] Moriah Ellen,[4] Anette Hulth,[5] Sonja Löfmark,[5] Gary E Garber,[6,7] William Watson,[8,9] Jacob Moran-Gilad,[10] David N Fisman,[1] Derek R MacFadden [1] [11]

DNF and DRM are joint senior authors.

For numbered affiliations see end of article.

**Correspondence to**
Dr Derek R MacFadden;
dmacfadden@toh.ca

## ABSTRACT

**Introduction** Antimicrobial resistance undermines our ability to treat bacterial infections, leading to longer hospital stays, increased morbidity and mortality, and a mounting burden to the healthcare system. Antimicrobial stewardship is increasingly important to safeguard the efficacy of existing drugs, as few new drugs are in the developmental pipeline. While significant progress has been made with respect to stewardship in hospitals, relatively little progress has been made in the primary care setting, where the majority of antimicrobials are prescribed. OPEN Stewardship is an international collaboration to develop an automated feedback platform to improve responsible antimicrobial prescribing among primary care physicians and capable of being deployed across heterogeneous healthcare settings. We describe the protocol for an evaluation of this automated feedback intervention with two main objectives: assessing changes in antimicrobial prescribing among participating physicians and determining the usability and usefulness of the reports.

**Methods and analysis** A non-randomised evaluation of the automated feedback intervention (OPEN Stewardship) will be conducted among approximately 150 primary care physicians recruited from Ontario, Canada and Southern Israel, based on a series of targeted stewardship messages sent using the platform. Using a controlled interrupted time-series analysis and multilevel negative binomial modelling, we will compare the antimicrobial prescribing rates of participants before and after the intervention, and also to the prescribing rates of non-participants (from the same healthcare network) during the same period. We will examine outcomes targeted by the stewardship messages, including prescribing for antimicrobials with duration longer than 7 days and prescribing for indications where antimicrobials are typically unnecessary. Participants will also complete a series of surveys to determine the usability and usefulness of the stewardship reports.

### Strengths and limitations of this study

► Multinational pragmatic study.
► Large number of providers studied with extended follow-up.
► Comprehensive evaluation including pre–post design and concurrent controls.
► Scalable and open concept with potential for broad generalisability to various care settings.
► Differences in recruitment strategy for intervention group across the two study sites may result in heterogeneity of intervention effect.

**Ethics and dissemination** All sites have obtained ethics committee approval to recruit providers and access anonymised prescribing data. Dissemination will occur through open-access publication, stakeholder networks and national/international meetings.

## INTRODUCTION

The WHO has called antimicrobial resistance 'one of the biggest threats to global health, food security and development today'.[1] Antimicrobial resistance undermines our ability to treat bacterial infections, leading to longer hospital stays, increased complications, higher mortality and a mounting resource burden on healthcare systems globally.[1–5] It has been projected that resistant microbes could contribute to up to 10 million deaths per year by 2050, greater than the current annual burden of cancer deaths. Additionally, a cumulative loss in global production of trillions of US dollars in the same time

frame is possible,[6] hindering efforts to eliminate poverty.[7] Despite a widespread recognition of antimicrobial use as the leading driver of resistance, inappropriate prescribing remains common.[8–10] Given the paucity of new antimicrobials in the drug development pipeline,[11] it is critical that we act to safeguard the effectiveness of existing drugs by reducing inappropriate antimicrobial prescribing.

Antimicrobial stewardship refers to the effort to promote the responsible use of antimicrobials and to ensure that patients receive the best choice, duration and route of antimicrobial treatment, while minimising potential side effects and the dissemination of antimicrobial resistance.[12] While progress has been made in advancing antimicrobial stewardship in hospitals, leading to significant cost savings and improved patient outcomes,[13 14] progress in the primary care setting has been slower.[15–17] This is troubling, given that most antimicrobials are prescribed in the outpatient setting, rather than in hospitals (eg, over 90% in Canada).[18 19] Supporting general and family practitioners with prescribing is especially important, since they account for the majority of antimicrobials dispensed for human use in Canada.[19] The US Centers for Disease Control and Prevention have issued a checklist identifying the key elements to a successful antimicrobial stewardship programme in an outpatient setting, highlighting easy access to evidence-based prescribing guidelines and the technical capacity to track, evaluate and report on antimicrobial prescribing.[20] An online, automated feedback system facilitating these elements would be a major asset to advancing stewardship in primary care.

OPEN Stewardship is an international collaboration with the goal of developing an open, web-based platform to improve antimicrobial prescribing among primary care physicians. This platform integrates with local antimicrobial prescribing data to provide participating physicians with regular, automated feedback (identified by a recent Cochrane review as an effective technique to enable appropriate antimicrobial prescribing in the hospital setting[14]) and connects users with locally relevant prescribing guidelines. The development of a standardised, accessible stewardship platform addresses a key gap for stewardship interventions in the primary care setting, namely the difficulty of expanding across heterogeneous, resource-constrained healthcare systems. As part of a One Health approach recognising the interconnectedness of human and animal health,[21] the platform is also being developed for use in veterinary care, a sector where antimicrobial stewardship has received less attention.[22] A study parallel to the present study will be occurring among veterinary care providers.

## STUDY APPROACH AND OBJECTIVES

In this study, we will enrol primary care physicians on a voluntary basis from practices in two study sites: Ontario, Canada and Southern Israel. Non-participating physicians (from the same healthcare network) will be available as controls. The intervention group will receive three personalised electronic reports sent over the span of 6 months, benchmarking their prescribing rate to those of their peers and attaching locally relevant prescribing guidelines for specific indications. The messaging will target antimicrobial prescribing longer than 7 days (few conditions require longer than this[23]), as well as two indications in particular: acute sinusitis and viral respiratory conditions. These common infections are a promising target for stewardship.[8 20 24 25] In the USA, acute respiratory conditions as a whole account for close to half of antimicrobials prescribed and a great majority of those prescribed inappropriately in the ambulatory care setting.[8]

The aim of this project is to assess the impact of an automated feedback intervention on antimicrobial prescribing in primary care physicians. Specifically, for a cohort of primary care physicians in Ontario, Canada and Southern Israel, we will determine the impact of the stewardship intervention on:

1. The overall antimicrobial prescribing rate in the period following the intervention compared with the period preceding it and to a cohort of primary care physicians who did not receive the intervention.
2. The prescribing rate for antimicrobials with duration longer than 7 days in the period following the intervention compared with the period preceding it and to a cohort of primary care physicians who did not receive the intervention.
3. The antimicrobial prescribing rate for viral respiratory conditions in the period following the intervention compared with the period preceding it and to a cohort of primary care physicians who did not receive the intervention.
4. The antimicrobial prescribing rate for acute sinusitis in the period following the intervention compared with the period preceding it and to a cohort of primary care physicians who did not receive the intervention.
   We will also assess:
5. The usability and usefulness of each component of the antimicrobial stewardship reports, as well as the usefulness of the reports overall.

## METHODS AND ANALYSIS
### Objectives 1–4: non-randomised evaluation of intervention with interrupted time-series analysis.
#### Study design
Primary care physicians (providers) from two study sites will be voluntarily enrolled to participate in an antimicrobial stewardship intervention. The providers' rates of antimicrobial prescribing after the intervention will be compared with their rates prior to having received the reports, as well as the prescribing rates of their peers in the same network who did not participate in the intervention. This study design is know as a controlled interrupted time-series.[26] Our study design addresses the limitations of some previous studies of stewardship interventions, including small sample sizes,

single-centre design, a lack of external controls and short follow-up.[16]

## Participants

Approximately 100–150 general practitioners will be enrolled from two study sites: Ontario, Canada (50) and Southern Israel (50–100). Participants will be drawn from the University of Toronto Practice-Based Research Network (UTOPIAN) and Clalit Health Services (CHS), respectively. UTOPIAN is 'one of the largest and most representative primary care research networks in North America',[27] including data for over 400 physicians with an average practice size of 1018 patients.[28] This includes family medicine practices and primary care practices in university-affiliated hospitals. CHS is the largest healthcare maintenance organisation in Southern Israel[29] and covers nearly 80% of the population in Southern Israel through a network family medicine practices. Existing data storage and sharing facilities within both of these networks allow access to de-identified patient data for research in the primary care setting.

In Ontario, primary care physicians will be voluntarily enrolled, provided they worked for at least 12 months prior to the intervention, expect to continue working for the 12 months following the start of the intervention and are not actively participating in another stewardship intervention. Providers will be approached for enrolment via email or in-person presentations to groups of providers. Participants will begin the intervention in at least two waves in different months. In Southern Israel, half of the approximately 400 eligible physicians (average number of patients: 1482) will be randomised (using random assignment in an Excel spreadsheet by the study coordinator) to be approached (via a letter from the chief physician, followed by contact by an interviewer) for voluntary enrolment in the study. Anonymised prescribing data of non-participating physicians within UTOPIAN and CHS will be available as an external control group (approximately 350 for UTOPIAN and 250–300 for CHS, with an expected 25%–50% response rate for physicians approached to enrol).

It is possible that physicians who voluntarily enrol in the antimicrobial stewardship intervention will be more engaged with stewardship prior to the intervention, and thus may have less to gain from a stewardship intervention than less-engaged physicians. Differences in the recruitment strategy between sites may result in a heterogeneity in the observed effect, which will be accounted for in the analysis.

## Intervention

The intervention will consist of three personalised reports generated by the OPEN Stewardship platform and emailed to participants in the intervention group. The first report will be sent at the beginning of the intervention period ($t_0$) and will consist of a welcome email followed by a benchmark of the provider's overall antimicrobial prescribing rate relative to the average and 25th percentile of participants in the same network (UTOPIAN or CHS) during the previous year (2019). This report will also include guidelines highlighting that most conditions do not require more than 7 days of antimicrobials and a benchmark of the provider's proportion of antimicrobial prescriptions with duration longer than 7 days relative to the average and 25th percentile of participants in the same network during the previous year. We chose to benchmark against other participants only (excluding non-participants) due to the expectation that those self-selecting into a stewardship intervention may already have lower rates of prescribing.

The two subsequent reports will be sent 3 ($t_3$) and 6 ($t_6$) months after the first report. These reports will target viral respiratory conditions (as defined by the primary International Classification of Diseases, Ninth Revision (ICD-9) code associated with the visit, see online supplemental table S1) and acute sinusitis, respectively, and will consist of a benchmark of the provider's prescribing for the specified indication as well as site-specific best practice guidelines for the specified indication. The study concludes 1 year after the first report is sent ($t_{12}$). The composition of each report is summarised in table 1. An example figure is shown in figure 1.

## Outcomes

Each of the four objectives corresponds to an outcome, measured for each provider as the monthly prescribing rate per 100 visits for a particular indication. This prescribing rate is calculated as the number of prescriptions for a particular indication divided by the number of patient visits for that indication in a month, multiplied by 100.

1. Overall prescribing rate for antimicrobials.
2. The prescribing rate for antimicrobials lasting longer than 7 days in duration.
3. The prescribing rate for visits classified as being for a viral respiratory condition (see online supplemental table S1 for ICD-9 codes).
4. The prescribing rate for acute sinusitis visits.

While overall antimicrobial prescribing is a proxy for our desired measure, which is appropriate antimicrobial prescribing, the microbiological data do not exist to validate the appropriateness of each individual prescription. Nonetheless, given the ubiquity of inappropriate prescribing, particularly for acute respiratory tract infections, any reduction in prescribing as a result of the intervention is likely to be positive.[8 30 31]

## Covariates

The analysis will include potential predictors of a provider's baseline prescribing rate, as well as seasonality (table 2). These include a provider's age, sex and employment status (full-time or part-time),[10] and monthly practice patient composition (by age and sex).[19 32] We will also use nested random intercepts for provider, practice group (clinic) and study site, representing within-group prescribing norms and a provider's baseline willingness

**Table 1** Composition of the feedback reports (intervention) at each time point

| Feedback report | Content of report | Benchmarking figure | Guidelines |
|---|---|---|---|
| Report at $t_0$ | 1. Antimicrobial prescribing rate/100 visits in the previous year for all visits.<br>2. Proportion of antimicrobial prescriptions in the previous year with duration over 7 days. | Provider's own rate/proportion versus average rate/proportion and 25th percentile rate/proportion across all participants at the same study site. | Guidelines indicating that antimicrobial prescribing longer than 7 days is not necessary for many conditions. |
| Report at $t_3$ | 1. Antimicrobial prescribing rate/100 visits in the previous year for viral respiratory conditions. | Provider's own rate versus average rate and 25th percentile rate across all participants at the same study site. | Prescribing guidelines for viral respiratory conditions. |
| Report at $t_6$ | 1. Antimicrobial prescribing rate/100 visits in the previous year for acute sinusitis. | Provider's own rate versus average rate and 25th percentile rate across all participants at the same study site. | Prescribing guidelines for acute sinusitis. |

to prescribe after accounting for other predictors, as well as heterogeneity between study sites. Seasonality will be accounted for using a categorical variable for each month compared with January. An interaction term with study site will be included for each covariate to allow for differences in seasonal and demographic prescribing patterns between the two countries in our study.

## Statistical analysis

We will perform a multilevel, controlled interrupted time-series analysis with the provider as the unit of analysis.[33] A negative binomial model (to allow for overdispersion

**Number of Prescriptions of Any Antibiotic (Per 100 Visits) for Any Indication**

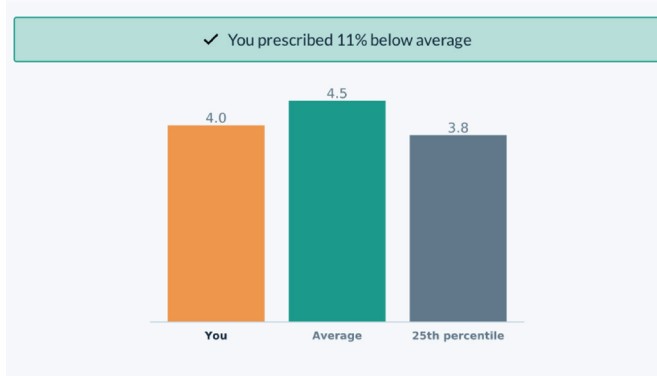

You had 5001 patient visits from Jan 01, 2019 to Dec 31, 2019.

This figure shows how frequently you prescribed antibiotics for any indication in the previous year (2019). Your prescribing rate (prescriptions per 100 visits) is compared to the average (mean) of your colleagues and participants in Southern Ontario as well as showing your position relative to the 25th percentile. Generally, it is better to be below the average rate of prescribing and within the lowest 25th percentile.

These data are not adjusted for physician's practice characteristics, nor do they differentiate between delayed or non-filled prescriptions. These results are generated from claims data, and misclassification of the diagnosis may be present. As a result, our prescribing targets are referenced to peer benchmarks. Learn more about the data and its interpretation.

**Figure 1** Example of chart sent in the first feedback report, representing the provider's prescribing rate/100 visits for any indication versus the average rate and 25th percentile rate across all providers in the study from the same study site.

of the outcome) will be used with prescribing rates as the outcome. Each provider in the intervention group will contribute 24 observations: a minimum of 12 months in the pre-intervention period ($t_{-12}$ to $t_0$) and up to 12 months in the post-intervention period, depending on the objective. The pre-intervention and post-intervention periods of each outcome are defined by the time when the relevant report was delivered. The first report ($t_0$) corresponds to objectives 1 and 2, the second report ($t_3$) to objective 3 and the third report ($t_6$) to objective 4. The pre-intervention and post-intervention periods of each objective are summarised in table 3. Providers in the external control group will contribute 24 observations in the control period. The timeline of the study for each objective is shown in figure 2.

The intervention effect will be modelled as a level change and/or change in slope for the prescribing rate at the beginning of the post-intervention period.[26] An interaction term with study site will be included to allow for differences in the intervention effect between Ontario and Southern Israel. A key assumption of interrupted time-series analysis is that the pre-intervention time trend is assumed to be linear, which must be evaluated through visualisation.[34] Otherwise, non-linear terms may be considered. Model fit can be assessed by comparing models containing intervention parameters to a null model not containing intervention parameters using a statistic such as Akaike information criterion.

Autocorrelation in each provider's time-series should be accounted for by the random intercept (provider nested within practice group nested within study site) and by controlling for seasonality (categorical variable for month).[26] In case autocorrelation is present in the residual distribution, an autoregressive covariance structure will be considered.

Stewardship interventions often result in a reversion to baseline behaviour after the intervention is discontinued.[35 36] To examine the durability of the intervention effect, we will contrast the coefficients estimated from

**Table 2** Covariates included in the interrupted time-series analysis of the OPEN Stewardship Study

| Variable | Purpose | Form | Description |
|---|---|---|---|
| Number of prescriptions | Outcome | Integer | Number of antimicrobial prescriptions for given indication |
| Number of visits for indication | Offset | Integer | Number of patient visits for given indication |
| Study site | Covariate | Binary | Southern Israel versus Ontario, Canada (baseline) |
| Practice group | Random intercept | ID | Random intercept for practice group (nested within region) |
| Provider | Random intercept | ID | Random intercept for provider (nested within practice group) |
| Month | Covariate | Categorical | Calendar month (baseline=January) |
| Month×study site | Interaction | Interaction | Allows different value for coefficients for Southern Israel |
| Physician age | Covariate | Integer | Physician age at $t_{-12}$ |
| Physician age×study site | Interaction | Interaction | Allows different value for coefficient for Southern Israel |
| Physician sex | Covariate | Binary | Female versus male (baseline) |
| Physician sex×study site | Interaction | Interaction | Allows different value for coefficient for Southern Israel |
| Physician employment status | Covariate | Binary | Part-time versus full-time (baseline) |
| Physician employment status×study site | Interaction | Interaction | Allows different value for coefficient for Southern Israel |
| Percent female patients | Covariate | Continuous | Percentage of visits by female patients |
| Percent female patients×study site | Interaction | Interaction | Allows different value for coefficient for Southern Israel |
| Percent patients 17 and under | Covariate | Continuous | Percentage of visits by patients 17 and younger |
| Percent patients 17 and under×study site | Interaction | Interaction | Allows different value for coefficient for Southern Israel |
| Percent patients 65 and older | Covariate | Continuous | Percentage of visits by patients 65 and older |
| Percent patients 65 and older×study site | Interaction | Interaction | Allows different value for coefficient for Southern Israel |
| Post-intervention | Intervention | Binary | 1 if provider is in intervention group and intervention has occurred, 0 otherwise; level change of the outcome associated with intervention |
| Post-intervention×study site | Interaction | Interaction | Allows different value for level change of the outcome for Southern Israel |
| Time | Trend | Integer | Time since the beginning of the study; slope of the time trend of the outcome |
| Time×study site | Interaction | Interaction | Allows different value for slope of the time trend of the outcome for Southern Israel |
| Time×post-intervention | Intervention effect | Interaction | Change in slope of time trend in the intervention group after intervention is received |
| Time×post-intervention×study site | Interaction | Interaction | Allows different value for change in slope of time trend in the intervention group after intervention is received for Southern Israel |

Excludes variables related to the effect of the COVID-19 pandemic.

**Table 3** Timing of analyses for objectives 1–4 of the study

| Objective | Outcome | Unit of analysis | Intervention time | Number of observations in intervention cohort | | Total observations/ observations in control cohort |
| | | | | Pre-intervention | Post-intervention | |
|---|---|---|---|---|---|---|
| 1 | Prescribing for all indications | Provider | $t_0$ | 12 | 12 | 24 |
| 2 | Prescribing (for all indications) with duration longer than 7 days | Provider | $t_0$ | 12 | 12 | 24 |
| 3 | Prescribing for viral respiratory conditions | Provider | $t_3$ | 15 | 9 | 24 |
| 4 | Prescribing for acute sinusitis | Provider | $t_6$ | 18 | 6 | 24 |

The definitions of the pre-intervention and post-intervention periods are dependent on the delivery time of the report (intervention) related to each objective.

models including (1) all months in the post-intervention period, (2) the first 3 months in the post-intervention period, (3) the first 6 months in the post-intervention period. Finally, since the first report at $t_0$ (highlighting overall prescribing) is likely to affect objectives targeted by later reports (at $t_3$ or $t_6$), for objectives 3 and 4, we will consider $t_0$ as an alternative start time of the post-intervention period.

As the COVID-19 pandemic represents a major disruption of primary care, its effect on prescribing will be modelled as one or more separate 'intervention points' in the time-series analysis.

### Sensitivity analyses

Alternative parameterisations of the study outcomes will be considered. For the prescribing rate of antimicrobials with courses longer than 7 days (objective 2), we will consider total days of therapy (with the number of prescriptions as the offset) as an alternative outcome using a zero-truncated negative binomial model. There may be issues with how indications are coded in the electronic medical record, whereas the type of antimicrobial prescribed is very likely to be accurate. A sensitivity analysis will be performed using the prescribing rates (for all visits) for antimicrobials most often prescribed for viral respiratory conditions (eg, amoxicillin, amoxicillin/clavulanic acid, azithromycin[37]) as the outcome, rather than the indication-specific prescribing rate of antimicrobials for viral respiratory conditions. Since acute respiratory tract infections make up a large proportion of total antimicrobial prescribing, the signal from the intervention may be evident using these outcomes as well.

To ascertain whether there could be negative effects of the stewardship intervention, we will evaluate (1) the frequency of bacterial pneumonia and pyelonephritis

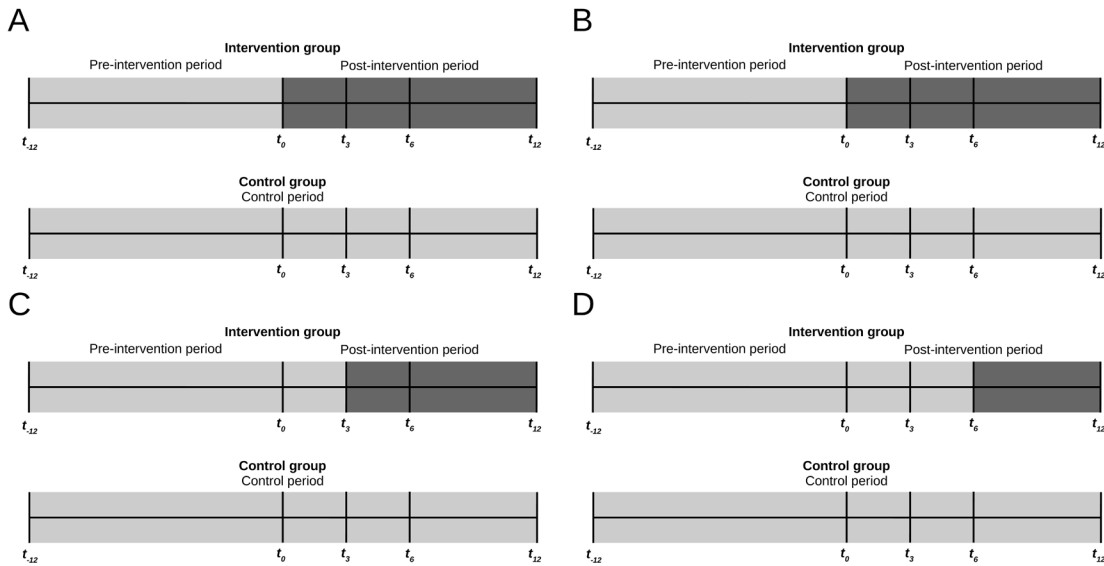

**Figure 2** Pre-intervention and post-intervention periods for objectives 1–4 of the OPEN Stewardship Study. (A) Objective 1, overall prescribing (report sent at $t_0$). (B) Objective 2, prescribing with duration longer than 7 days (report sent at $t_0$). (C) Objective 3, prescribing for viral respiratory conditions (report sent at $t_3$). (D) Objective 4, prescribing for acute sinusitis (report sent at $t_6$). $t_0$ represents the time of the first report (intervention); $t_{-12}$ represents 1 year prior to the first report; $t_{12}$ marks the conclusion of the study period (1 year after the first report).

diagnoses and (2) the number of hospital admissions (including those related to infection) in the pre-intervention and post-intervention periods. Admissions data are only available for CHS (Israel), and thus will only be evaluated at this site only.

Including patient composition as predictors (eg, percentage of patients 65 and older, percentage of female patients) may be insufficient to capture differences in prescribing occurring between strata of age and sex. A sensitivity analysis will be performed where the outcome will be stratified by age group and sex and modelled separately. Although stratification increases the noise in each individual time-series, these models may reveal differences in the effectiveness of the intervention across sex/age groups, which have different prescribing patterns.

As the offer to enrol in the study is randomised at one study site (Southern Israel), an intention-to-treat analysis will be performed as a supplementary analysis at this site only. The characteristics and prescribing patterns of responders and non-responders among those offered enrolment will also be compared.

### Sample size

We provide an approximate sample size calculation for a difference in antimicrobial prescribing between two groups (intervention and control) in a Poisson framework using the method described by Whitehead[38] and implemented in an online calculator (http://www.obg.cuhk.edu.hk/ResearchSupport/StatTools/SSiz2Counts_Tab.php). In Fleming-Dutra *et al's*[8] study of outpatient prescribing, 12.6% of visits resulted in an antimicrobial prescription. Since most visits would only result in a single antimicrobial prescription, we take this as the baseline prescribing rate (0.126/visit). The American Academy of Family Physicians estimated in 2012 that their members had an average of 92 patient encounters per week (368 per month).[39] We define a 15% reduction in prescribing rate as clinically meaningful (0.107/visit). To achieve a power of 90% (with $\alpha=0.05$, one-tail), we would require approximately 15 physicians in each group to be observed for 1 month each, which is much smaller than our expected sample size at each study site. While the clustering and likely overdispersion of prescribing rates in our sample would decrease our power, the fact that we will have 24 months of observations for each provider (rather than 1 month), as well as pre-intervention and post-intervention observations, greatly enhances the power of our study to detect an intervention effect.

### Objective 5: survey of usability among participating providers
#### Study design

Physicians enrolled in the antimicrobial stewardship intervention will receive three short surveys to assess the usability of the reports received and to provide feedback on the intervention.

### Surveys

Surveys will be delivered by email to participants in the intervention group 3 days after the initial receipt of each report using Qualtrics (Qualtrics, Provo, Utah, USA) in Southern Israel and RedCAP[40] survey software in Ontario (due to differences in institutional policies regarding data collection). The report will be resent with the survey and the recipient will be given 7 days to respond. If the recipient fails to respond within this allotted time, the survey and report are resent again, giving the recipient an additional 7 days to respond. The first two surveys focus on the usability and usefulness of the emailed reports, with additional space for free text feedback. The final report adds additional questions to assess how often they would prefer to receive reports and whether particular additional features would be of use to him or her. Survey questions are given in English in online supplemental appendix 1; translated questions in Hebrew are given in online supplemental appendix 2. Questions were developed in accordance with experts at the Public Health Agency of Sweden. Questions were pilot tested and revised according to the protocol and interviewer guide in online supplemental appendix 3.

### Statistical analysis

Participants' attitudes regarding the overall usefulness of the intervention and the informativeness of the figure included in each intervention (table 1) will be assessed in each survey and compared longitudinally with a multilevel linear model. Responses to other questions may be analysed using ordinal regression and/or multinomial regression.

## ETHICS AND DISSEMINATION

The following ethics committees reviewed and approved this study: University of Toronto Research Ethics Board (REB) (Canada), St. Michael's Hospital REB (Canada), Toronto East Health Network REB (Canada) and Clalit REB (Israel). The veterinary study was further reviewed and approved by: University of Guelph REB (Canada) and Internal Research Review Committee of the Koret School of Veterinary Medicine–Veterinary Teaching Hospital, Hebrew University (Israel).

All patient-level prescribing data will be anonymised prior to receipt, available only as the rate of antimicrobial prescribing per visit, stratified by month and reason for visit (and potentially sex and/or broad age category). Study data will be stored on a secure server and accessible only to study analysts. Study participants from Ontario will be compensated with a gift card. Study participants from Israel will be encouraged to participate by a letter from the chief physician's office.

The OPEN Stewardship project includes collaborators affiliated with hospitals, universities and public health authorities in four countries. In addition to dissemination through stakeholder networks, the results of this study

will be published in an open-access journal and presented at national and/or international meetings.

## Patient and public involvement

Primary care physicians were involved in the conception and design of this study, as well as the study surveys. They were also consulted to assess the appropriate time burden and adequate compensation for participants in the study.

**Author affiliations**
[1]Division of Epidemiology, Dalla Lana School of Public Health, University of Toronto, Toronto, Ontario, Canada
[2]Chief Physician's Office, Clalit Health Services, Tel Aviv, Israel
[3]Department of Population Medicine, University of Guelph Ontario Veterinary College, Guelph, Ontario, Canada
[4]Department of Health Services Management, Guilford Glazer Faculty of Business and Management, University of the Negev, Beer Sheva, Israel
[5]The Public Health Agency of Sweden, Stockholm, Sweden
[6]Infection Prevention and Control, Public Health Ontario, Toronto, Ontario, Canada
[7]Infectious Diseases, Faculty of Medicine, University of Ottawa, Ottawa, Ontario, Canada
[8]Department of Family & Community Medicine, University of Toronto, Toronto, Ontario, Canada
[9]St Michael's Hospital, Toronto, Ontario, Canada
[10]School of Public Health, Faculty of Health Sciences, Ben-Gurion University of the Negev, Beer Sheva, Israel
[11]Clinical Epidemiology Program, The Ottawa Hospital Research Institute, Ottawa, Ontario, Canada

**Contributors**  J-PRS, ML, KRA, ME, AH, SL, GEG, WW, JM-G, DNF and DRM conceived the study and the analysis. J-PRS wrote the initial manuscript and will conduct the data analysis. All authors contributed to and approved the final manuscript.

**Funding**  This work was supported by the European Commission's Joint Programming Initiative on Antimicrobial Resistance (Swedish Research Council: 2017-05972, Canadian Institutes of Health Research: AMR-155212, Israeli Ministry of Health: 8762491).

**Competing interests**  None declared.

**Patient consent for publication**  Not required.

**Provenance and peer review**  Not commissioned; externally peer reviewed.

**ORCID iDs**
Jean-Paul R Soucy http://orcid.org/0000-0002-8422-2326
Kamal Raj Acharya http://orcid.org/0000-0001-6707-3536
Derek R MacFadden http://orcid.org/0000-0002-3838-1211

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
