## [Reviewer comments · BMJ Open]

ARTICLE DETAILS

TITLE (PROVISIONAL)	Evaluation of an automated feedback intervention to improve antimicrobial prescribing among primary care physicians (OPEN Stewardship): Protocol for an interrupted time-series and usability analysis in Ontario, Canada and Southern Israel
AUTHORS	Soucy, Jean-Paul; Low, Marcelo; Acharya, Kamal; Ellen, Moriah; Hulth, Anette; Löfmark, Sonja; Garber, Gary; Watson, William; Moran-Gilad, Jacob; Fisman, David N.; MacFadden, Derek

VERSION 1 – REVIEW

REVIEWER	Prof Dave Brodbelt Royal Veterinary College, UK
REVIEW RETURNED	20-Jul-2020

GENERAL COMMENTS	General This is an interesting study that is likely to yield valuable results. The main concern relates to the reliability of evaluating prescribing levels for the two clinical conditions stated, as it would appear currently these conditions do not appear sufficiently defined to be confident AM usage would relate to a single clearly defined condition. Further and related to this it would at present remain uncertain whether there is sufficient evidence within the EPR to assess the reliability of diagnoses made for these two conditions to be confident they are unlikely to include bacterial disease. This part of the protocol would need further consideration and details. Abstract. Line 20 onwards – methods – it would be helpful to include reference to the statistical approaches in the abstract also. Introduction Page 4 Line 45-46. It may well be the case veterinary AM stewardship has received less attention but this would need a reference to support this. Lines 48 onwards – this section would appear more appropriate in the methods. Consider moving this section or reword to focus on the previous work only rather than the methods Page 5 – lines 3-16- move to the methods also. Aims and objectives appear appropriate. Objective 4 – it will be important to clarify how these conditions are confirmed as viral based on the data available. Methods Page 5 Line 23 – it is not clear why all centres were not to receive an email and then the investigators follow up in a random order the eligible physicians until the target recruitment is achieved.
--

	Line 28 – is there evidence to support this assumed response rate? Please clarify. Page 6 – Line 26 onwards – given the study has the data it would be valuable also to evaluate the non-responder prescriber levels compared to the non-recruited external controls. Both groups are in the external control group but it would be valuable to assess if non-responders appeared different in characteristics and or usage to those that were not approached beyond an introductory email. This is commented on also in the analysis section Line 48 – it would be helpful to provide clarification of the case definitions and inclusion/exclusion for acute sinusitis and viral respiratory conditions here. There is some reference to this in the outcomes but it would be appropriate to clarify this here. Page 7 Line 14 – for clarity it would be helpful to define provider either here or where first referred to – e.g. where the target number of physicians is stated in the methods. Line 15. Some measures of the provider’s practice location socio-economic status would also be valuable here. Statistical analysis Line 27 onwards – it would be helpful to state the assumptions of the models and how these will be assessed as well as report measure of model fit that the investigators plan to use to evaluate the models reported. Line 37 – it would be useful to also compare provider characteristics and AM usage between non-responders and those not approached within the external control group in a similar approach to the proposed comparison of the intervention to control groups. Line 55 – it would be beneficial to describe the likely autocorrelation covariance structures to be considered and the basis for such a decision. Page 8 Line 2 – clarify how the effect of covid 19 will be assessed Line 9 – the evaluation of mean duration may not be appropriate if the duration outcome is skewed. This is quite likely – hence a non-parametric summary measure of outcome duration may be more appropriate. Unless the authors have prior evidence suggesting a normal distribution of duration over 7 days. Page 9 Statistical analysis Given there is likely to be a number of participants with incomplete data – i.e. response to some but not all surveys. It may be more appropriate to consider a mixed effects linear model here over a repeated measures anova. Ethics Line 36 onwards. it would be valuable to have the ethics approval documentation in the methods also. i.e. if approval has been granted then the ethics approval IDs could be included here. Line 42 – would it be possible to have comparable incentives across sites? This might reduce heterogeneity in the study.
--	---

REVIEWER	Thulasiraman P Faculty of Veterinary and Animal Sciences, Institute of Agricultural Sciences, Rajiv Gandhi South Campus, Banaras Hindu University
REVIEW RETURNED	24-Jul-2020

GENERAL COMMENTS	The study design, methods employed and the outcomes were described sufficiently. The need of the hour study.
--

REVIEWER	Laurel Redding University of Pennsylvania
-----------------	--

REVIEW RETURNED	31-Jul-2020
-------------

GENERAL COMMENTS	Overall I think this study is a really great idea and desperately needed in veterinary medicine, I just have some concerns about how you are proposing to carry it out, especially in terms of the metrics used and the choice of data to include. Comparing antimicrobial prescribing rates for all visits among peers may engender defensive behavior or even rejection of the reports among the participants if they do not think the comparison is fair, and I think it may be very difficult to establish that the peers to which vets will be compared are sufficiently comparable. How can you ensure comparability of the patient populations seen by the veterinarians? Even matching by practice type may not be sufficient. As far as I know, the antimicrobial stewardship initiatives using feedback in human medicine provide reporting on specific disease conditions for which there are unambiguous guidelines (e.g., upper respiratory disease), not overall prescribing for all patients seen, which lends itself to better credibility by veterinarians. If you peruse the human medicine literature, you will see that prescribing clinician resistance to feedback data is a critical barrier to these being an effective stewardship initiative (e.g., Szymczak et al. 2014, Pediatrician perceptions of an outpatient antimicrobial stewardship intervention, Infect Control Hosp Epidemiol). How will you deal with this issue? I presume the idea with these reports is to get veterinarians to just be thinking about their prescribing in general, but I worry about a potential backlash against – or arguably, worse, indifference to these reports. As someone who has performed a pilot version of what you propose to do among small and large animal veterinarians and specifically captured qualitative feedback on these reports and specific metrics (publication forthcoming in Frontiers), I can tell you that veterinarians may become defensive or feel judged by these reports comparing them to their peers if they do not perceive comparability of patient populations among their peers and if there is a lack of evidence based guidelines on which to base judiciousness of prescribing. That being said, the initiative is needed, and we do need to start somewhere, and you have a great proposed foundation here. We obviously do not want perfect to be the enemy of the good, but I hope you will consider some of the points I suggest below and perhaps do a more thorough review of the qualitative literature associated with this topic to help refine your proposal. L 170 – I think you mean we will enroll primary care vets on a voluntary basis. L180-182 and line 256: I do not see how your reference 45 is relevant here; I also am very surprised that you would consider most treatments for mastitis inappropriate and unnecessary. Do you have any evidence for this claim? L206: It is unclear whether this study will be conducted with the context of a lengthy and inclusive national or clinic-level discussion of the positive contribution of AMU benchmarking to broader efforts towards reduced AMU. Will the veterinarians enrolled have any introduction to this study and to antimicrobial stewardship in general, or will they be getting these reports cold? L217-218: why such a large discrepancy between the number of vets in the control and intervention practices?
---

	L207-216: will these be outpatient clinics exclusively? Private practice exclusively? General practice only? Will antibiotic prescribing include inpatients and outpatients, or outpatients only? You've divided up your practices by species, but what about by type of service (e.g., specialty vs emergency vs GP)? Obviously the patient population seen by each type of practice will vary substantially, so comparisons may not be deemed entirely fair. L219-220: will the control veterinarians be from the same practice as the intervention veterinarians? If so, how can you protect against contamination of the control population? Two veterinarians working at the same practice, especially if it is a small clinic, will be likely to talk about the results given to the veterinarian with an intervention. Why not make the practice the unit of intervention (e.g., all veterinarians within a practice get the intervention and in control practices, none of the vets do?). L223: Please provide more detail on how the OPEN stewardship platform works. Do all of these practices have similar enough electronic medical record systems that enable the automatization of generating feedback reports? If so, which systems do they use and how is the prescribing information extracted? L224: Which metrics are to be used? Is it just # of visits where an antibiotic was prescribed per 100 visits? Or will there be more targeted metrics by diagnosis? L235: what is this guideline, and from what is it derived? L246: again, will the prescribing be detailed for overall or for specific disease conditions? Will the report have prescribing rates broken down into disease conditions? L247-250: how will you extract duration of therapy from the electronic medical records? Is it automatically included in each prescription? What about on dairy farms where farmers may be following a general treatment protocol rather than having an individual veterinarian administer/prescribe treatments? L249-250: which guidelines are these? There are very few conditions that have evidence-based guidelines recommending fewer than 7 days of therapy in veterinary medicine (beyond UTIs and URIs), so I'm wondering where you are getting these recommendations. L252: how do you define broad spectrum prescribing? Many of the broad spectrum antimicrobial agents are critically important antimicrobials, so I suspect that you will have a great deal of overlap with your first report. Also, how will you know if an etiological agent has or has not been identified? In many cases where the etiological agent is unknown, it is not inappropriate to prescribe a broad spectrum agent, so I am concerned that by providing a report on broad spectrum antibiotics without any clinical context or related diagnosis, you are sending a message that it is wrong to use these antimicrobials. L257: this really seems to be the type of information you would want to use qualitative methods to obtain. I wonder if a survey will really capture the nuances of how veterinarians feel about this report. L287: will there be any obligation to participate? Do they have to respond to the questionnaire to continue to be involved in the study? If not, what is your expected return rate? L294: do you mean a likert scale? L323: I presume you are using a negative binomial regression model because your outcome is counts of antibiotics prescribed? How does that not account for the denominator of total # of cases seen? If it is not a count-based model, what is the unit of your rates?
--	--

	L335-337: other studies have calculated baseline rates for antibiotic prescribing among veterinarians. Why not use some of these values for your power calculations? See  - Redding, L. E., et al. (2020). "Antimicrobial prescribing patterns of clinicians and clinical services at a large animal veterinary teaching hospital." Am J Vet Res 81(2): 103-115; - Singleton DA, et al. . "New approaches to pharmacosurveillance for monitoring prescription frequency, diversity, and coprescription in a large sentinel network of companion animal veterinary practices in the United Kingdom, 2014 to 2016". Preventive Veterinary Medicine (2018), https://doi.org/10.1016/j.prevetmed.2018.09.004 - Hughes LA, Williams N, Clegg P, et al. Cross-sectional survey of antimicrobial prescribing patterns in UK small animal veterinary practice. Prev Vet Med. 2012;104(3-4):309-316. doi:10.1016/j.prevetmed.2011.12.003 - Hughes LA, Pinchbeck G, Callaby R, Dawson S, Clegg P, Williams N. Antimicrobial prescribing practice in UK equine veterinary practice. Equine Vet J. 2013;45(2):141-147. doi:10.1111/j.2042-3306.2012.00602.x - Lutz B, Lehner C, Schmitt K, et al. Antimicrobial prescriptions and adherence to prudent use guidelines for selected canine diseases in Switzerland in 2016. Vet Rec Open. 2020;7(1):e000370. Published 2020 Mar 9. doi:10.1136/vetreco-2019-000370 - Schmitt K, Lehner C, Schuller S, et al. Antimicrobial use for selected diseases in cats in Switzerland. BMC Vet Res. 2019;15(1):94. Published 2019 Mar 14. doi:10.1186/s12917-019-1821-0 - Robbins SN, Goggs R, Lhermie G, Lalonde-Paul DF, Menard J. Antimicrobial Prescribing Practices in Small Animal Emergency and Critical Care. Front Vet Sci. 2020;7:110. Published 2020 Feb 28. doi:10.3389/fvets.2020.00110 - Both Banfield reports - https://www.banfield.com/Banfield/media/contenthub/files/2018_VET_Report.pdf and https://www.banfield.com/getmedia/e6c50f42-9ded-4323-aa03-4fd92a2aa012/VET-Report_Final_web.pdf - L351-355: more information on metrics and on the units and denominators of these metrics is needed. L364-365: you may also want to consider years in practice and place of education, as I am sure those will have significant impacts on exposure to stewardship measures.
--	--

VERSION 1 – AUTHOR RESPONSE

Reviewer 1

General

This is an interesting study that is likely to yield valuable results. The main concern relates to the reliability of evaluating prescribing levels for the two clinical conditions stated, as it would appear

currently these conditions do not appear sufficiently defined to be confident AM usage would relate to a single clearly defined condition. Further and related to this it would at present remain uncertain whether there is sufficient evidence within the EPR to assess the reliability of diagnoses made for these two conditions to be confident they are unlikely to include bacterial disease. This part of the protocol would need further consideration and details.

We appreciate the constructive feedback provided by this reviewer. We have addressed the reviewer's specific concerns below.

Abstract

Line 20 onwards – methods – it would be helpful to include reference to the statistical approaches in the abstract also.

We have added a reference to the statistical approach in the abstract:

“Using a controlled interrupted time-series analysis and multilevel negative binomial modelling”

Introduction

Page 4 / Line 45-46. It may well be the case veterinary AM stewardship has received less attention but this would need a reference to support this.

A reference to a book chapter discussing how the “development of AMS in veterinary medicine has lagged behind that in the human medicine field” (Lloyd & Page, 2018) has been added to support this statement.

Lines 48 onwards – this section would appear more appropriate in the methods. Consider moving this section or reword to focus on the previous work only rather than the methods

Page 5 – lines 3-16- move to the methods also.

We agree that this content would be more appropriate in the methods section. The first paragraph has been moved to introduce the aims and objectives section. The second paragraph has been integrated into the study design and outcomes sections of the methods, where appropriate.

Aims and objectives

Aims and objectives appear appropriate.

Objective 4 – it will be important to clarify how these conditions are confirmed as viral based on the data available.

We cannot microbiologically confirm that the conditions for which these antimicrobials have been prescribed are viral in every case. As a proxy, we use ICD-9 codes (see table S1) corresponding to indications that are most likely viral in nature and where antimicrobials are not routinely recommended. This has been clarified in the expanded outcomes section:

“While overall antimicrobial prescribing is a proxy for our desired measure, which is appropriate antimicrobial prescribing, the microbiological data do not exist to validate the appropriateness of each individual prescription. Nonetheless, given the ubiquity of inappropriate prescribing, particularly for acute respiratory tract infections, any reduction in prescribing as a result of the intervention is likely to be positive.”

This section cites the appropriate literature justifying the selection of ICD-9 codes, particularly Fleming-Dutra et al. (2016).

Methods

Page 6 / Line 23 – it is not clear why all centres were not to receive an email and then the investigators follow up in a random order the eligible physicians until the target recruitment is achieved.

The investigators in Israel were interested in being able to perform an intention-to-treat analysis as a sensitivity analysis, which could only be achieved by randomizing the offer to participate. This is described in the sensitivity analysis section.

Line 28 – is there evidence to support this assumed response rate? Please clarify.

The assumed response rate is based on the experience of the investigators in Israel. We have clarified this in the manuscript:

“with an expected 25–50% response rate for physicians approached to enrol”

Page 6/Line 26 onwards – given the study has the data it would be valuable also to evaluate the non-responder prescriber levels compared to the non-recruited external controls. Both groups are in the external control group but it would be valuable to assess if non-responders appeared different in characteristics and or usage to those that were not approached beyond an introductory email. This is commented on also in the analysis section

This would make an excellent supplement to the existing sensitivity analysis at the Israel site (intention-to-treat analysis of those offered enrolment versus those not offered enrolment). We have added the following:

“As the offer to enrol in the study is randomized at one study site (Southern Israel), an intention-to-treat analysis will be performed as a supplementary analysis at this site only. **The characteristics and prescribing patterns of responders and non-responders among those offered enrolment will also be compared.”**

Line 48 – it would be helpful to provide clarification of the case definitions and inclusion/exclusion for acute sinusitis and viral respiratory conditions here. There is some reference to this in the outcomes but it would be appropriate to clarify this here.

The conditions are defined based on ICD-9 codes attached to each visit. We have added the following line to the intervention section to clarify this:

“These reports will target acute sinusitis and viral respiratory conditions (as defined by the ICD-9 code associated with the visit, see table S1)”

Page 7/Line 14 – for clarity it would be helpful to define provider either here or where first referred to – e.g. where the target number of physicians is stated in the methods.

Provider has now been defined at the first place it is mentioned (the beginning of the study approach and objectives section).

Line 15. Some measures of the provider’s practice location socio-economic status would also be valuable here.

Unfortunately, we are receiving data in an anonymized form based on already-existing data held by our data partners, so we will be unable to add this variable. This will be mentioned in the discussion of the analysis.

Statistical analysis

Line 27 onwards – it would be helpful to state the assumptions of the models and how these will be assessed as well as report measure of model fit that the investigators plan to use to evaluate the models reported.

This section has been expanded to include a key assumption of interrupted time series analysis as well as how AIC will be used to evaluate model fit with the intervention parameters:

“A key assumption of interrupted time-series analysis is that the pre-intervention time trend is assumed to be linear, which must be evaluated through visualization. Otherwise, non-linear terms may be considered. Model fit can be assessed by comparing models containing intervention parameters to a null model not containing intervention parameters using a statistic such as Akaike information criterion (AIC).”

Line 37 – it would be useful to also compare provider characteristics and AM usage between non-responders and those not approached within the external control group in a similar approach to the proposed comparison of the intervention to control groups.

Agreed. See response to related comment made in the methods section.

Line 55 – it would be beneficial to describe the likely autocorrelation covariance structures to be considered and the basis for such a decision.

We have clarified that the autocorrelation covariance structure would be AR(1) if autocorrelation is present in the residual distribution.

Page 8/Line 2 – clarify how the effect of covid 19 will be assessed

We have added the following line to clarify:

“As the COVID-19 pandemic represents a major disruption of primary care, its effect on prescribing will be modelled as one or more separate “intervention points” in the time-series analysis.”

Line 9 – the evaluation of mean duration may not be appropriate if the duration outcome is skewed. This is quite likely – hence a non-parametric summary measure of outcome duration may be more appropriate. Unless the authors have prior evidence suggesting a normal distribution of duration over 7 days.

This is a valid concern. A count-based regression model based on total days of therapy would be more appropriate. Thus, we have replaced this section with:

“For the prescribing rate of antimicrobials with courses longer than seven days (objective 2), **we will consider total days of therapy (with the number of prescriptions as the offset) as an alternative outcome using a zero-truncated negative binomial model.**”

Page 9 – Given there is likely to be a number of participants with incomplete data – i.e. response to some but not all surveys. It may be more appropriate to consider a mixed effects linear model here over a repeated measures anova.

We agree this is likely to be the case. We have modified the line:

“Participants’ attitudes regarding the overall usefulness of the intervention and the informativeness of the figure included in each intervention (table 1) will be assessed in each survey and compared longitudinally **with a multilevel linear model.**”

Ethics

Line 36 onwards. it would be valuable to have the ethics approval documentation in the methods also. i.e. if approval has been granted then the ethics approval IDs could be included here.

As requested by the editor, we have added the list of ethics committees that granted approval for this study in this section.

Line 42 – would it be possible to have comparable incentives across sites? This might reduce heterogeneity in the study.

Unfortunately, the incentives are set by the investigators at each site based on the particular circumstances and healthcare systems of the sites.

Reviewer 2

The study design, methods employed and the outcomes were described sufficiently. The need of the hour study.

We are pleased the author enjoyed our protocol.

Reviewer 3 (commenting on our sister manuscript ID bmjopen-2020-039760)

L257: this really seems to be the type of information you would want to use qualitative methods to obtain. I wonder if a survey will really capture the nuances of how veterinarians feel about this report.

This comment refers to the surveys to evaluate usability and feedback, which will be performed in both the human and veterinary studies. We agree that qualitative feedback would be helpful, which is why our final survey asks “Would you be willing to participate in a follow-up interview where we would ask in more detail about your experiences with the OPEN Stewardship reports? If so, please send an email to [study email address].” (Appendix 1)

VERSION 2 – REVIEW

REVIEWER	Prof Dave Brodbelt Royal Veterinary College, UK
REVIEW RETURNED	08-Sep-2020

GENERAL COMMENTS	The authors have addressed the reviewer's queries.
--

REVIEWER	Laurel Redding University of Pennsylvania
REVIEW RETURNED	19-Sep-2020

GENERAL COMMENTS	Apologies for having submitted the review for your companion article on veterinarians. Please find below my review for this article. I reviewed the revised article, not the original article. Overall, this is a really neat study idea and very much needed. The study plan is well designed, the methods are clear for the most part, and I think these studies will significantly advance the field of antimicrobial stewardship in both veterinary and human medicine. I do wonder how the COVID pandemic will affect the study, especially given the respiratory manifestations of the disease. Will physicians be more or less likely to be prescribing antibiotics in the general time of COVID? Or if a patient is specifically suspected of
---

	having COVID? Could you integrate whether or not a covid test was ordered as a covariate? Otherwise, please see specific comments below. On a side note, please use continuous lines that line up exactly with your text. P4, L 52: How will non-participating physicians be enrolled? Will they have to consent to having their data used? How do you plan to address the selection bias if you are using self-selecting participants? P4, L54: Which peers? Within their practice? Their country? All participants in the study? P5, L53...: A bit of background on the types of primary care practices these are would be appreciated. Do they see children and adults? Adults only? Are they family medicine practices, urgent care centers, affiliated with university settings? Privately owned? Will there be a fair amount of heterogeneity in the type of practice enrolled? How will you ensure that the directors of all of the clinics approached for participation buy-in to the study? P6, L7-19: will you be comparing the physicians who agreed and were randomized to the intervention and those who agreed and were not randomized to the intervention? Or are you comparing those who agreed to the non-participating controls? A diagram may be helpful in understanding your randomization/enrollment scheme. Also, will you have any way to control against contamination of your control group, for example, if one of your participants in the intervention is in the same practice as a non-participant/control and could share information on the intervention with the non-participant? LP6, L9-10: what will determine whether you approach patients by email or in-person presentations? And who will give the in-person presentations? I imagine you will have very different response rates based on mode of recruitment. P6, L 34-36: will the non-participating physicians also have access to the guidelines? Otherwise, how will you differentiate the effect of the guidelines vs the peer comparison? P6, L39: why use ICD-9 and not ICD-10 codes? P 6, L51-56. What are the units of metrics 2-4? For metric 1, you note here that the metric is defined as, “overall monthly prescribing rate for antimicrobials, as measured by the number of prescriptions divided by the number of patients visits in that month”, whereas in Table 1, you note that the unit of the metric is “Antimicrobial prescribing rate/100 visits in the previous year”. Also, will there be some overlap in the cases listed in metrics 3 and 4? I.e., could a physician code a visit as acute sinusitis and acute viral infection? Related to that, if multiple diagnoses are listed, which one do you choose to use? Only the primary diagnosis? Or do you consider other ones as well? Is there some sort of threshold for how many of the icd-9 codes need to meet your criteria? For example, if code 786.2 “cough” is one of the diagnoses, but some sort of cardiovascular diagnosis is also
--	---

	present that could indicate congestive heart failure, who makes the decision to include that visit or not? P7, L9. You may also want to consider length of time in practice as a covariate, as I suspect that will affect prescribing practices. P8, L31-33: will you have any information on socioeconomic status of patients or co-morbidities of patients? Or is sex and age the only covariates you will have access to? Figure 1: The metric I am most familiar with is percent of visits where an antimicrobial is prescribed. Does this come out to be the same thing? If so, why not just label it “percent of visits where you prescribed an antimicrobial”? “Per 100 visits” sounds a bit abstract and could potentially be confusing. Figure 2: Do you anticipate any issues with the fact that the comparison time period will be different for control groups (24 m) vs the intervention groups (12-18 months), especially in light of the covid pandemic?
--	--

VERSION 2 – AUTHOR RESPONSE

Review 1

The authors have addressed the reviewer's queries.

We are grateful for the feedback this reviewer provided.

Reviewer 3

General

Overall, this is a really neat study idea and very much needed. The study plan is well designed, the methods are clear for the most part, and I think these studies will significantly advance the field of antimicrobial stewardship in both veterinary and human medicine.

I do wonder how the COVID pandemic will affect the study, especially given the respiratory manifestations of the disease. Will physicians be more or less likely to be prescribing antibiotics in the general time of COVID? Or if a patient is specifically suspected of having COVID?

We agree this is a major concern, which is why we have decided to incorporate the effect of the COVID-19 pandemic in our interrupted time-series analysis. This is described in the manuscript:

“As the COVID-19 pandemic represents a major disruption of primary care, its effect on prescribing will be modelled as one or more separate “intervention points” in the time-series analysis.”

Could you integrate whether or not a covid test was ordered as a covariate?

Unfortunately, these data are not available at our study sites. They would also be of limited value since data are analyzed at the aggregate rather than individual level, so we would only be able to look at the percentage of patients in a month referred for a COVID-19 test. We expect to capture the effects of COVID-19 on prescribing by including it in the time series analysis, as mentioned previously. Since we include both an intervention group and non-participating controls, it will be possible to describe the trend in prescribing attributable to changes in healthcare induced by the pandemic.

Specific comments

P4, L 52: How will non-participating physicians be enrolled? Will they have to consent to having their data used? How do you plan to address the selection bias if you are using self-selecting participants?

Physicians are being recruited from healthcare networks with existing data use agreements (University of Toronto Practice-Based Research Network in Ontario and Clalit Health Services in Southern Israel). As described in the manuscript, anonymized prescribing data from non-participating physicians within these are available to be used as a control group:

“Existing data storage and sharing facilities within both of these networks allow access to de-identified patient data for research in the primary care setting. [...] Anonymized prescribing data of non-participating physicians within UTOPIAN and CHS will be available as an external control group.”

We agree that self-selection is a concern with intervention studies and have highlighted this fact in the manuscript:

“It is possible that physicians who voluntarily enrol in the antimicrobial stewardship intervention will be more engaged with stewardship prior to the intervention, and thus may have less to gain from a stewardship intervention than less-engaged physicians.”

However, our choice of study design (controlled interrupted time-series analysis) addresses this concern by analyzing the relative change in prescribing in the same providers before and after the intervention while controlling for external time trends through the incorporation of a control group.

P4, L54: Which peers? Within their practice? Their country? All participants in the study?

Provider are benchmarked against other participants in the study (i.e., the intervention group) at the same study site (Ontario, Canada or Southern Israel). We describe this in the intervention section:

“The first report will be sent at the beginning of the intervention period (t_0) and will consist of a welcome email followed by a benchmark of the provider’s overall antimicrobial prescribing rate relative to the average and 25th percentile of participants in the same network (UTOPIAN or CHS) during the previous year (2019).”

We have also added a line to this section to give the rationale as to why only participants in the study are included in the benchmark (as opposed to participants and non-participants in the same network). This pertains to the self-selection bias mentioned earlier:

“We chose to benchmark against other participants only (excluding non-participants) due to the expectation that those self-selecting into a stewardship intervention may already have lower rates of prescribing.”

P5, L53....: A bit of background on the types of primary care practices these are would be appreciated. Do they see children and adults? Adults only? Are they family medicine practices, urgent care centers, affiliated with university settings? Privately owned? Will there be a fair amount of heterogeneity in the type of practice enrolled? How will you ensure that the directors of all of the clinics approached for participation buy-in to the study?

Participants are general practitioners. In Ontario, Canada we will draw from a mix of family medicine practices and primary care practices in university-affiliated hospitals, whereas in Southern Israel we will draw mainly from family medicine practices. We have added some of this context to the participants section:

“Approximately 100–150 general practitioners will be enrolled from two study sites: Ontario, Canada (50) and Southern Israel (50–100). Participants will be drawn from the University of Toronto Practice-Based Research Network (UTOPIAN) and Clalit Health Services (CHS), respectively. UTOPIAN is “one of the largest and most representative primary care research networks in North America”, including data for over 400 physicians with an average practice size of 1,018 patients. This includes family medicine practices and primary care practices in university-affiliated hospitals. CHS is the largest healthcare maintenance organization in Southern Israel and covers nearly 80% of the population in Southern Israel through a network family medicine practices.”

We acknowledge the differences between the two sites may create heterogeneity, which we have built into our statistical analysis (specific variables are described in Table 2):

“Differences in the recruitment strategy between sites may result in a heterogeneity in the observed effect, which will be accounted for in the analysis.”

We do not expect clinic buy-in to be an issue beyond the mechanism of recruitment (which will be a function of recruitment mechanisms already in place), since each physician operates with autonomy.

P6, L7-19: will you be comparing the physicians who agreed and were randomized to the intervention and those who agreed and were not randomized to the intervention? Or are you comparing those who agreed to the non-participating controls? A diagram may be helpful in understanding your randomization/enrollment scheme. Also, will you have any way to control against contamination of your control group, for example, if one of your participants in the intervention is in the same practice as a non-participant/control and could share information on the intervention with the non-participant?

Since physicians are recruited individually (rather than an entire clinic), contamination is a possibility in our study, which could dilute the measured treatment effect. However, sparking discussions with peers is something we have considered as a possible use for the automated stewardship reports, which is why we ask about this in our survey (Survey 1, Q5, Rate each of the following items according to its potential usefulness to you in an OPEN Stewardship report: “Have discussions with peers on the local situation regarding prescribing patterns”). Whether participants shared their report with others would be a good question to ask in the qualitative interviews with a subset of participants that are planned after the intervention is concluded.

Randomization to be contacted for voluntary recruitment will occur only at the Southern Israel site, due to the centralized nature of the participating healthcare network (CHS) and preference of the site PI. For the purposes of the primary analysis, this is simply a difference in recruitment mechanism compared to the other site (Ontario, Canada). In both sites, participating physicians will be compared to themselves pre-intervention as well to non-participating physicians (whether they were randomized to be asked to voluntarily enrol or not). However, in the Southern Israel site, we are capable of performing a sensitivity analysis due to the randomization mechanism:

“As the offer to enrol in the study is randomized at one study site (Southern Israel), an intention-to-treat analysis will be performed as a supplementary analysis at this site only. The characteristics and prescribing patterns of responders and non-responders among those offered enrolment will also be compared.”

LP6, L9-10: what will determine whether you approach patients by email or in-person presentations? And who will give the in-person presentations? I imagine you will have very different response rates based on mode of recruitment.

The recruiting mechanism is a function of the clinic, their existing engagement in research initiatives, and the preferences of the site PIs. While these different mechanisms may produce different response rates, we do not expect self-selection to be an issue for the analysis since providers are being compared against themselves (pre-intervention) as well as external controls.

P6, L 34-36: will the non-participating physicians also have access to the guidelines? Otherwise, how will you differentiate the effect of the guidelines vs the peer comparison?

Our objective is to look at the effect of the overall intervention (the automated stewardship report), not disentangle the effects of guidelines versus peer comparison. The guidelines themselves are not novel, but our approach of presenting them alongside peer benchmarks is.

P6, L39: why use ICD-9 and not ICD-10 codes?

The databases of the participating healthcare networks in Ontario, Canada and Southern Israel use ICD-9 codes, so this aspect of the study is beyond our control.

P 6, L51-56. What are the units of metrics 2-4? For metric 1, you note here that the metric is defined as, "overall monthly prescribing rate for antimicrobials, as measured by the number of prescriptions divided by the number of patients visits in that month", whereas in Table 1, you note that the unit of the metric is "Antimicrobial prescribing rate/100 visits in the previous year".

We have modified the outcomes section to make the units for the outcomes more clear:

"Each of the 4 objectives corresponds to an outcome, measured for each provider as the monthly prescribing rate per 100 visits for a particular indication. This prescribing rate is calculated as the number of prescriptions for a particular indication divided by the number of patient visits for that indication in a month, multiplied by 100."

The intervention section describes the units given in Table 1, which are the prescribing rate as described above except using the provider's data from the entire previous year:

"a benchmark of the provider's overall antimicrobial prescribing rate relative to the average and 25th percentile of participants in the same network (UTOPIAN or CHS) during the previous year (2019)"

Also, will there be some overlap in the cases listed in metrics 3 and 4? I.e., could a physician code a visit as acute sinusitis and acute viral infection? Related to that, if multiple diagnoses are listed, which one do you choose to use? Only the primary diagnosis? Or do you consider other ones as well? Is there some sort of threshold for how many of the icd-9 codes need to meet your criteria? For example, if code 786.2 "cough" is one of the diagnoses, but some sort of cardiovascular diagnosis is also present that could indicate congestive heart failure, who makes the decision to include that visit or not?

The databases of the participating healthcare networks in Ontario, Canada and Southern Israel record only a single ICD-9 associated with each visit, the primary diagnosis. We have amended the intervention section slightly to clarify this:

"The two subsequent reports will be sent 3 (t_3) and 6 (t_6) months after the first report. These reports will target viral respiratory conditions (as defined by the primary ICD-9 code associated with the visit, see table S1)"

P7, L9. You may also want to consider length of time in practice as a covariate, as I suspect that will affect prescribing practices.

We agree this would be a useful covariate. However, birth year and sex are unfortunately the only covariates available to us regarding the personal characteristics of the providers. We expect birth year to correlate with length of time in practice.

P8, L31-33: will you have any information on socioeconomic status of patients or co-morbidities of patients? Or is sex and age the only covariates you will have access to?

We agree socio-demographic or comorbidity covariates would be potentially useful. However, age and sex are the only patient-level covariates available, and they can only be included in aggregate (e.g., percent of visits in a month from patients that are 65+). Since a provider's patients should be based on the composition of the surrounding community, we expect the practice-level random intercepts to account for differences in patient populations to some extent, since general practitioners at the same practice should see a similar patient population.

Figure 1: The metric I am most familiar with is percent of visits where an antimicrobial is prescribed. Does this come out to be the same thing? If so, why not just label it "percent of visits where you prescribed an antimicrobial"? "Per 100 visits" sounds a bit abstract and could potentially be confusing.

Our decision to present results as per 100 visits rather % of visits was driven by our discussions with physicians when designing our figures, who felt that the former was a more tangible metric than the latter. This format also allows for the possibility of prescribing multiple antibiotics for a single visit.

Figure 2: Do you anticipate any issues with the fact that the comparison time period will be different for control groups (24 m) vs the intervention groups (12-18 months), especially in light of the covid pandemic?

Total follow-up time should be the same for both groups (24 months total). For all outcomes, at least 12 pre-intervention months are included for the intervention group. This pre-intervention period will consist of time before and after significant changes related to the COVID-19 pandemic (March 2020). This means the effect of the COVID-19 pandemic can be captured in both the control and intervention groups.

VERSION 3 – REVIEW

REVIEWER	Laurel Redding University of Pennsylvania
REVIEW RETURNED	16-Nov-2020
GENERAL COMMENTS	Thank you for addressing my concerns. Best of luck with the study!